# The Beneficial Roles of Elevated [CO₂] on Exogenous ABA-Enhanced Drought Tolerance of Cucumber Seedlings

Qiying Sun [1], Xinrui He [1], Tengqi Wang [1], Hengshan Qin [1], Xin Yuan [2], Yunke Chen [2], Zhonghua Bian [2,*] and Qingming Li [2,*]

1  College of Horticulture Science and Engineering, Shandong Agricultural University, Tai'an 271018, China
2  Institute of Urban Agriculture, Chinese Academy of Agricultural Sciences, National Chengdu Agricultural Science and Technology Center, Chengdu 610299, China
*  Correspondence: bianzhonghua@caas.cn (Z.B.); liqingming@caas.cn (Q.L.)

**Abstract:** Drought stress severely limits crop growth and yield. With the atmospheric $CO_2$ constantly increasing, plants will be affected by multiple effects of drought and increased $CO_2$ in the future. Abscisic acid (ABA) plays vital roles in plant stress tolerance, especially drought stress. However, little is known about the effects of elevated $CO_2$ concentration (e[$CO_2$]) and exogenous ABA in cucumber (*Cucumis sativus* L.) response to drought stress. In the present study, we investigated the effects of e[$CO_2$] and exogenous ABA on the drought tolerance of cucumber seedlings under the simulated drought stress induced by 5% polyethylene glycol 6000. The experiment was a split-plot design, in which the main factor was $CO_2$ concentrations; atmospheric and elevated $CO_2$ concentrations (~400 and 800 ± 40 µmol mol$^{-1}$, respectively). The subplot factor was the combinations of exogenous ABA and its synthesis inhibitor sodium tungstate (Na$_2$WO$_4$); deionized water (control), 20 µM ABA, 2 mM Na$_2$WO$_4$, and 2 mM Na$_2$WO$_4$ + 20 µM ABA, which were applied to plant leaves. The results showed that compared with exogenous ABA application only, e[$CO_2$] combined with exogenous ABA significantly increased the biomass, chlorophyll content, and net photosynthetic rate ($P_n$) of cucumber seedlings under drought stress. Meanwhile, e[$CO_2$] and exogenous ABA were more efficient in reducing the contents of reactive oxygen species and malondialdehyde, promoting the accumulation of proline, soluble sugar, soluble protein, free amino acid, ascorbic acid, and glutathione. The ratios of ascorbic acid/dehydroascorbic acid (ASA/DHA), glutathione/oxidized glutathione (GSH/GSSG), as well as the activities of antioxidant enzymes were increased. In conclusion, e[$CO_2$] and exogenous ABA synergistically alleviated oxidative damage of drought stress on cucumber seedlings by increasing antioxidant enzyme activities and accelerating the ASA–GSH cycle in cucumber seedlings, which in turn improved the drought tolerance of cucumber seedlings, and provided theoretical and practical support for further studies on the alleviation of drought stress in protected horticulture.

**Keywords:** drought stress; elevated $CO_2$; ABA; antioxidant system; ASA–GSH

## 1. Introduction

Atmospheric $CO_2$ concentrations have increased dramatically from the industrial revolution to the present and are expected to reach 600 µmol mol$^{-1}$ by mid-century [1]. As atmospheric $CO_2$ concentrations have increased, the frequency of droughts and high temperatures have increased worldwide, which in turn can lead to significant impacts on crop quality and yields [2,3]. This is because drought reduces the water potential of soil solution, induces oxidative stress which hampers many biological processes, causes osmotic stress, imbalance of free radical metabolism, membrane lipid peroxidation, and other damage to plant cells [4,5]. Meanwhile, when plants are subjected to drought stress, plant growth is limited and leads to reduced chlorophyll content, inhibited photosynthesis, reduced dry matter accumulation, and causes a reduction in crop growth and productivity [6].





Oxidative stress of plants was induced by the generation of reactive oxygen species (ROS) which include hydrogen peroxide ($H_2O_2$), superoxide anion ($O_2^-$), and oxygen-containing free radicals. The production and clearance of ROS in healthy plants maintains a dynamic balance. However, drought stress could destroy this dynamic balance, which leads to rising levels of ROS and thus hampers the growth of plants [7,8]. Plants have evolved a set of effective antioxidant defense systems to eliminate ROS [9], involving enzymatic and non-enzymatic antioxidant systems [10]. The enzymatic antioxidant system consists of superoxide peroxidase (POD), dismutase (SOD), and catalase (CAT). Furthermore, the abnormal accumulation of $H_2O_2$ can damage the membrane systems of tissues and organs, and subsequently the metabolic activities [11]. The ascorbate–glutathione (ASA–GSH) cycle is also an important way to remove $H_2O_2$ in plants [12,13], mainly in the cytoplasm and chloroplasts. ASA and GSH are two important antioxidants, which can scavenge reactive oxygen species and maintain the redox state of plants under various stresses [14–16] and their existing forms also affect their functions. When subjected to a certain degree of oxidative damage, plants can convert oxidized ascorbic acid and oxidized glutathione into their corresponding reduced state to reduce the damage [12,17]. Moreover, accumulation of osmotic proteins is a non-enzymatic antioxidant defense system to improve tissue water status by osmoregulation and to quench plant ROS under stress conditions.

For crops, $CO_2$ enrichment can promote plant growth by improving the photosynthetic efficiency of plants [18]. Recent studies showed that elevated $CO_2$ concentration (e[$CO_2$]) could mitigate some deleterious effects in coffee plants at the physiological and metabolic level [19], and increase sugar accumulation of *Populus* leaves and upregulation of genes determining anthocyanin biosynthesis to prolong leaf longevity during natural autumnal senescence [20]. Moreover, e[$CO_2$] plays an important role in alleviating the damage caused by abiotic stresses such as drought, salt, high temperature, etc. [21,22]. It can increase the activity of antioxidant enzymes like SOD, POD, CAT, ascorbate peroxidase (APX), and glutathione reductase (GR), and reduce the accumulation of reactive ROS and the degree of membrane lipid peroxidation [23], thereby improving plant resistance and alleviating oxidative damage caused by drought stress [24,25].

Abscisic acid (ABA) is known as the "stress hormone" of plants, and can alleviate the damage to plants in drought and other adverse events and improve the resistance of plants [26]. Plants release ABA in the roots, and then transfer it to the leaves through xylem fluid movement, resulting in a large accumulation of ABA content in the leaves [27]. The abundance of ABA reduces transpiration and improves water use efficiency by promoting stomatal closure [25,28]. Meanwhile, ABA works as the transmitter to trigger the response of plants to adversity stress, which improves the activities of related antioxidant enzymes to some extent in plants, thereby reducing the accumulation of reactive ROS in plants [29] and inducing plant tolerance to stress [30–32]. It was reported that the ASA–GSH cycle under ABA could reduce growth inhibition through larger antioxidant capacity, lower levels of ROS, and stronger tolerance to drought stress [11]. Thus, many drought-resistant varieties are more sensitive to ABA, while drought-susceptible varieties are the opposite [33]. Previous studies proved that the application of exogenous ABA could alleviate the oxidative damage to kiwifruit and soybean under drought stress, and enhance their drought resistance [34,35].

Recently, the effects of e[$CO_2$] in the regulation of plant drought response to ABA were reported [36,37]. Chater et al. [38] showed that ABA could regulate the decrease in stomatal density caused by the increase in $CO_2$ concentration. In addition, ABA could induce stomata closure under e[$CO_2$] [39,40], which means that ABA could interact with $CO_2$ to regulate the stomatal opening and closing and consequently facilitate plants to adapt to drought conditions. At present, there are many studies that have focused on the roles of $CO_2$ and ABA in drought stress, but most of them have focused on stomatal movement and less on active oxygen metabolism and the ASA–GSH cycle which are important reducing substances in plants.

This study aims to explore the effects of e[$CO_2$] and ABA in relieving oxidative damage of cucumber seedlings under drought stress. We hypothesized that e[$CO_2$] and exogenous ABA would enhance the reactive oxygen metabolism and ASA cycling in cucumber seedlings under drought conditions, which would improve the drought tolerance of cucumber seedlings. The results of this research could not only provide useful information for enhancing crop drought tolerance but also could give guidance on high-yield production in the greenhouse.

## 2. Materials and Methods

### 2.1. Plant Materials and Treatments

This experiment was conducted in four self-designed open-top tunnels (6 m length, 6 m span, and 2.6 m ridge height) at the Horticulture Experimental Station of Shandong Agricultural University, China. Cucumber (*Cucumis sativus* L. cv. Jinyou No. 35; from Tianjin Kerun Cucumber Research Institute, Tianjin, China) seeds were washed with distilled water and then soaked in warm water for 6–8 h, and later incubated for 1.5 d in a dark environment with a relative humidity of 60–70% and temperature of 28 °C. Then, cucumber seeds were sown in 50-plug black plastic trays (one seed per plug containing a mixture of peat, perlite, and vermiculite with a 3:1:1 volume ratio). After the first true leaves appeared and the roots were carefully washed, all the seedlings were transplanted into the solution culture system and planted in 7 L containers filled with Yamazaki cucumber nutrient solution. Each container had six seedlings, and there were 16 such containers in each self-designed open-top tunnel. The nutrient solution was replaced every three days and aerated with an air pump for 30 s every three minutes. The pH of the nutrient solution was maintained at 6.8–7.0 and the contents of the solution included 0.5 mM $NH_4H_2PO_4$, 2.0 mM $Ca(NO_3)_2 \cdot 4H_2O$, 3.2 mM $KNO_3$, 1.0 mM $MgSO_4 \cdot 7H_2O$, and full-strength trace elements.

The experiment was a split-plot design, in which the main factor was $CO_2$ concentration: atmospheric $CO_2$ concentration (a[$CO_2$]; ~400 µmol mol$^{-1}$) and elevated $CO_2$ concentration (e[$CO_2$]; 800 ± 40 µmol mol$^{-1}$), and the subplot factor was the combinations of exogenous ABA and its synthesis inhibitor sodium tungstate ($Na_2WO_4$): deionized water (control), 20 µM ABA, 2 mM $Na_2WO_4$, and 20 µM ABA + 2 mM $Na_2WO_4$. The experiment was started when the second true leaf was fully expanded. For the first two days of the experiment, leaves of cucumber seedlings were, respectively, sprayed with deionized water (control), 20 µM ABA, 2 mM $Na_2WO_4$, and 20 µM ABA + 2 mM $Na_2WO_4$ twice a day (8:00 and 18:00), which inferred that plants in every four containers were sprayed with the same solution in the same tunnel. On the third day of the experiment, all the nutrient solutions were replaced with a nutrient solution containing 5% polyethylene glycol 6000 (PEG 6000) for simulating soil drought stress, which lasted for 5 days. According to the equation proposed by Michel and Kaufmann (1973) [41], the level of osmotic pressure in the hydroponic solution was −0.05 MPa when the PEG 6000 stress was 5%. On the third day of the experiment, a[$CO_2$] and e[$CO_2$] treatments were performed in different tunnels for 5 days, and $CO_2$ was automatically released into the tunnels to maintain the target concentration. Both open-top tunnels were equipped the environmental control system (Auto 2000; Beijing, China) to provide $CO_2$ via compressed $CO_2$ cylinders under solenoid valve control. The experiment with eight treatments was repeated three times, in four containers with six seedlings each in one treatment. After the seventh day of treatment, leaves of cucumber seedlings in each treatment were sampled, then immediately frozen in liquid nitrogen, and stored at −80 °C fridge for physiological analyses.

### 2.2. Determination of Growth Parameters

The morphological parameters of seedlings were recorded immediately after 5 days of drought treatment, including plant height, diameter thickness, leaf area, fresh weight and dry weight of the whole plant. Plant height was measured using a ruler. The stem diameter was measured one centimeter below the cotyledons using a digital caliper. Leaf

area was calculated using the formula $S_L = L_L{}^2$, with $S_L$ representing the leaf area and $L_L$ representing the leaf length of the true leaves. The fresh and dry weights were determined by the weighing method by first weighing the shoot and root fresh weights of the plants separately, then drying them in an oven at 80 °C to a constant weight and weighing the dry weights.

### 2.3. Measurement of Photosynthetic Pigment Content and Gas Exchange

Chlorophyll a, chlorophyll b, and carotenoid contents of cucumber leaves were extracted according to the 95% ethanol extraction method (1 sheet) by grinding 0.5 g of fresh leaf samples in a mortar and adding 10 mL of 95% ethanol. The samples were left in the dark at 4 °C for 48 h and then centrifuged at $8000 \times g$ for 15 min. The absorbance values of the supernatants were measured at 665 nm, 649 nm and 470 nm using a spectrophotometer. The contents of chlorophyll a, chlorophyll b, and carotenoids were calculated according to the formulae of Lichtenthaler and Wellburn (1983) [42].

Photosynthetic gas exchange parameters were measured using a CIRAS-3 (PP Systems, US) portable plant photosynthesis meter for net photosynthetic rate ($P_n$), stomatal conductance ($G_s$), transpiration rate ($T_r$), and water use efficiency (WUE = $P_n / T_r$). Measurement conditions: leaf chamber temperature was 25 °C, light intensity was 1000 µmol m$^{-2}$ s$^{-1}$, and $CO_2$ was measured using buffer bottles taking a relatively stable 3~4 m high air at a concentration of about $400 \pm 10$ µmol mol$^{-1}$.

### 2.4. Measurement of MDA and ROS Contents

The content of malondialdehyde (MDA) was determined using the thiobarbituric acid (TBA) colorimetric method [43]. The $O_2{}^-$ generation rate was determined by the method of Elstner and Heupel [44]. The content of $H_2O_2$ was determined using reagent kits (Comin Biotechnology Co., Ltd., Suzhou, China) according to the instructions of the manufacturer (http://www.cominbio.com/a/shijihe/shenghuashiji/yanghuayukangyanghuaxilie/2016/0422/877.html) accessed on 14 February 2021.

### 2.5. Measurement of Osmolytes' Contents

The content of proline was determined according to the ninhydrin coloration method [45]. Determination of free amino acid content was by the ninhydrin method [46]. The contents of soluble sugars were determined by the anthrone method, and the contents of soluble proteins were determined by the Coomassie Brilliant Blue G-250 method [47].

### 2.6. Determination of Leaf ASA/DHA and GSH/GSSG

The content of ascorbate (ASA) was estimated according to the method proposed by [48]. The content of reduced glutathione (GSH) was determined by colorimetry of 5, 5-dithiobis (2-nitrobenzoic acid) (DTNB) [49]. Dehydroascorbate (DHA) and oxidized glutathione (GSSG) contents were determined using reagent kits (Comin Biotechnology Co., Ltd., Suzhou, China) according to the manufacturer's instructions. The content of DHA was calculated by measuring the rate of ASA generation in the system. The content of GSSG was appraised by abolishing GSH with a derivatizing agent 2-vinyl pyridine (http://www.cominbio.com/a/shijihe/shenghuashiji/guguanggantaixilie/2014/0226/212.html) accessed on 14 February 2021.

### 2.7. Enzyme Assays

Superoxidase dismutase (SOD) enzyme activity was assessed by measuring the inhibition of the photochemical reduction of nitroblue tetrazolium (NBT) at 560 nm [50]; peroxidase (POD) enzyme activity was measured by the guaiacol method [51]; catalase (CAT) enzyme activity was analyzed by measuring the decline in absorbance at 240 nm.

Ascorbate peroxidase (APX) activity was measured according to the method of [52]. Glutathione reductase (GR), glutathione peroxidase (GPX), and dehydroascorbate reductase (DHAR) activities were determined using reagent kits (Comin Biotechnology Co., Ltd.,

Suzhou, China) according to the instructions of the manufacturer (http://www.cominbio.com/a/shijihe/shenghuashiji/guguanggantaixilie/) accessed on 14 February 2020.

*2.8. Statistical Analyses*

Statistical analyses were carried out by Microsoft Excel 2019 (Microsoft Corporation, Redmond, WA, USA) and DPS 15.10 (Zhejiang University, Hangzhou, China). Data were presented as the means of six seedlings in each treatment with three repetitions. Split-plot analyses of variances were applied to evaluate the interactions between $CO_2$ and the combinations of ABA and $Na_2WO_4$. Duncan's multiple polar difference test ($\alpha = 0.05$) was used for all data to test for the presence of significant differences between treatments. Graphics were plotted with SigmaPlot 12.5 (Systat Software Inc, San Jose, CA, USA).

## 3. Results

*3.1. Effects of Exogenous ABA and Elevated $CO_2$ on Growth Parameters of Cucumber Seedlings under Drought Stress*

The results showed that e[$CO_2$] significantly promoted the growth of cucumber seedlings under drought stress compared with a[$CO_2$] (Figure 1), which specifically increased plant height, stem thickness, leaf area, and dry as well as fresh weights to different degrees, except for the $Na_2WO_4$ treatment (Table 1). Likewise, under a[$CO_2$], the plant fresh as well as dry weights and stem thickness of cucumbers in the ABA treatment were significantly higher than those of the control treatment, increasing by 7%, 14%, and 22%, respectively. The plant growth under the $Na_2WO_4$ treatment was the lowest compared to other treatments. Additionally, all growth parameters of the $Na_2WO_4$ + ABA treatment were significantly higher than those in the $Na_2WO_4$ treatment, while ABA spraying under e[$CO_2$] was more beneficial to plant growth under drought stress compared with other treatments (Table 1, Figure 1).

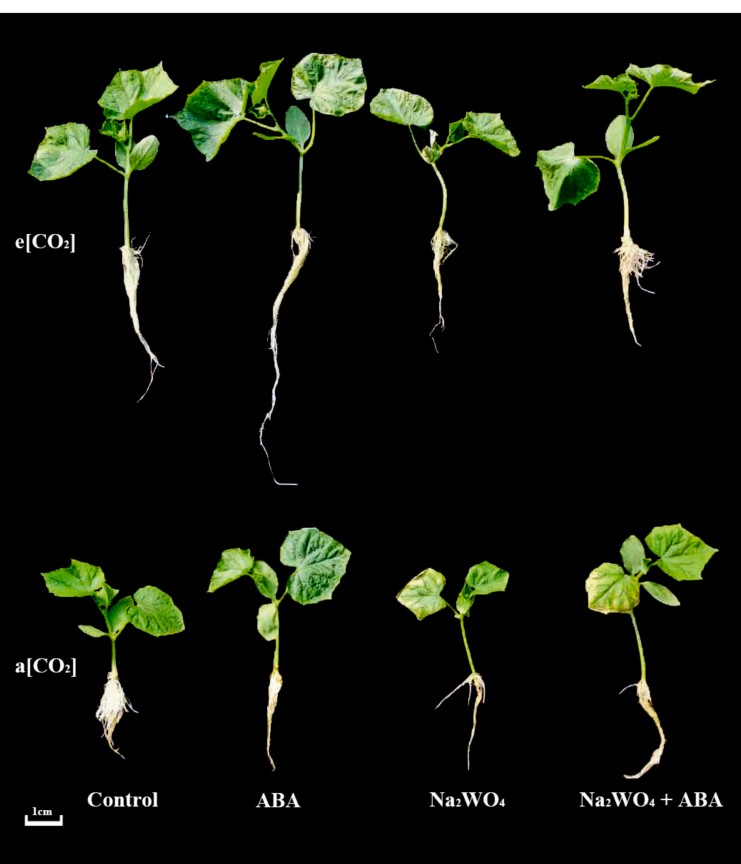

**Figure 1.** Side view of cucumber seedlings on day 5 under different treatments.

**Table 1.** Effects of elevated $CO_2$ and exogenous ABA on the growth of cucumber seedlings under drought stress.

| $CO_2$ Concentration | Treatments | Plant Height (cm) | Stem Thickness (cm) | Leaf Area (cm$^2$) | Fresh Weight (g) | Dry Weight (g) |
|---|---|---|---|---|---|---|
| a[$CO_2$] | control | 14.73 ± 0.17 c | 5.463 ± 0.105 ef | 355.27 ± 6.46 c | 21.35 ± 0.80 c | 2.41 ± 0.07 c |
| | ABA | 14.40 ± 0.14 c | 5.848 ± 0.192 cd | 368.78 ± 4.11 c | 24.39 ± 0.42 b | 2.95 ± 0.07 b |
| | Na$_2$WO$_4$ | 9.65 ± 0.68 e | 4.988 ± 0.133 g | 226.10 ± 6.59 e | 13.09 ± 0.44 f | 1.22 ± 0.05 e |
| | Na$_2$WO$_4$ + ABA | 13.52 ± 0.28 d | 5.605 ± 0.204 de | 360.73 ± 10.95 c | 18.66 ± 0.62 d | 2.13 ± 0.13 d |
| e[$CO_2$] | control | 18.78 ± 0.57 a | 6.183 ± 0.141 bc | 470.49 ± 11.87 a | 24.55 ± 0.53 b | 3.05 ± 0.09 b |
| | ABA | 18.48 ± 0.24 a | 6.777 ± 0.282 a | 439.99 ± 18.52 b | 30.96 ± 1.46 a | 4.18 ± 0.18 a |
| | Na$_2$WO$_4$ | 12.97 ± 0.41 d | 5.145 ± 0.080 fg | 271.80 ± 13.61 d | 15.23 ± 0.58 e | 1.48 ± 0.03 e |
| | Na$_2$WO$_4$ + ABA | 17.77 ± 0.4 b | 6.301 ± 0.136 b | 467.63 ± 6.03 a | 23.86 ± 0.23 b | 2.61 ± 0.08 c |

The data are means ± SEs (n = 6). The different letters in the same column indicate significant differences among treatments at $p < 0.05$.

### 3.2. Effects of Exogenous ABA and Elevated $CO_2$ on Photosynthetic Properties of Cucumber Seedling Leaves under Drought Stress

Under e[$CO_2$], exogenous ABA significantly increases the contents of chlorophyll a (Chl a), chlorophyll b (Chl b), and total chlorophyll (Total Chl), by 15%, 16%, and 15%, respectively, compared with the control treatment, but did not significantly affect the contents of chlorophyll a/b (Chl a/Chl b) and carotenoids (Table 2). On the other hand, under a[$CO_2$], Na$_2$WO$_4$ treatment showed the lowest content of all pigments compared with control treatment, and the differences between the levels of Chl b and carotenoid content were significant compared with the control group, but exogenous ABA spraying also did not significantly promote the accumulation of photosynthetic pigment content in leaves under drought stress.

**Table 2.** Effects of elevated $CO_2$ and exogenous ABA on photosynthetic pigment content of cucumber seedlings under drought stress.

| $CO_2$ Concentration | Treatments | Chl a (mg·g$^{-1}$ FW) | Chl b (mg·g$^{-1}$ FW) | Total Chl Content (mg·g$^{-1}$ FW) | Chl a/Chl b | Carotenoid (mg·g$^{-1}$ FW) |
|---|---|---|---|---|---|---|
| a[$CO_2$] | control | 1.334 ± 0.018 cd | 0.447 ± 0.010 d | 1.781 ± 0.017 cd | 2.983 ± 0.091 ab. | 0.344 ± 0.009 bc |
| | ABA | 1.489 ± 0.032 bc | 0.501 ± 0.009 cd | 1.991 ± 0.034 bc | 2.972 ± 0.086 ab | 0.381 ± 0.013 abc |
| | Na$_2$WO$_4$ | 1.148 ± 0.087 d | 0.376 ± 0.020 e | 1.524 ± 0.107 d | 3.050 ± 0.074 a | 0.253 ± 0.033 d |
| | Na$_2$WO$_4$ + ABA | 1.329 ± 0.050 cd | 0.441 ± 0.031 d | 1.769 ± 0.081 cd | 3.018 ± 0.108 ab | 0.311 ± 0.028 cd |
| e[$CO_2$] | control | 1.584 ± 0.011 b | 0.533 ± 0.001 b | 2.117 ± 0.010 b | 2.972 ± 0.027 ab | 0.392 ± 0.005 ab |
| | ABA | 1.828 ± 0.253 a | 0.616 ± 0.094 a | 2.444 ± 0.347 a | 2.975 ± 0.051 ab | 0.465 ± 0.059 a |
| | Na$_2$WO$_4$ | 1.647 ± 0.008 ab | 0.568 ± 0.006 ab | 2.214 ± 0.014 ab | 2.900 ± 0.018 ab | 0.411 ± 0.002 ab |
| | Na$_2$WO$_4$ + ABA | 1.461 ± 0.037 bc | 0.514 ± 0.017 bc | 1.975 ± 0.053 bc | 2.842 ± 0.028 b | 0.352 ± 0.010 bc |

The data are means ± SEs (n = 6). The different letters in the same column indicate significant differences among treatments at $p < 0.05$.

In control and ABA treatments, e[$CO_2$] significantly increased $P_n$ and WUE in drought stressed cucumber plants compared with a[$CO_2$] (Figure 2A,C). Under e[$CO_2$] and a[$CO_2$], Na$_2$WO$_4$ treatment resulted in the lowest $P_n$ and WUE compared with the control treatment. However, exogenous ABA spraying significantly enhanced the $P_n$ and WUE in leaves under e[$CO_2$]. Under a[$CO_2$], $P_n$ was significantly increased by 66% with ABA treatment and decreased by 30% with Na$_2$WO$_4$ treatment compared with control treatment, and increased by 84% with Na$_2$WO$_4$ + ABA treatment compared with Na$_2$WO$_4$ treatment. e[$CO_2$] treatment reduced plant $G_s$ and $T_r$ compared to those under a[$CO_2$], but this reduction was not significant under Na$_2$WO$_4$ treatment (Figure 2B,D).

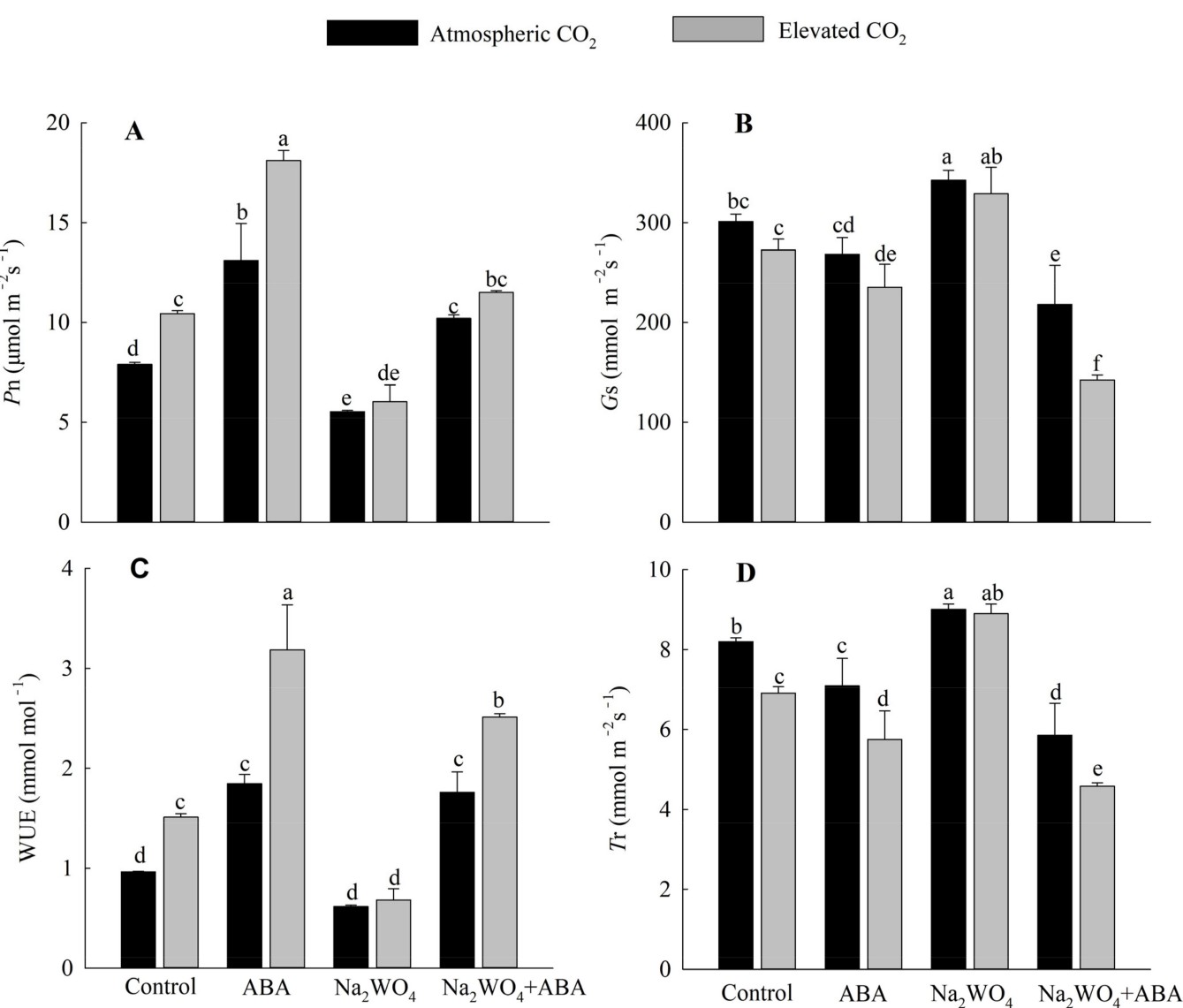

**Figure 2.** Effects of elevated $CO_2$ and exogenous ABA on net photosynthetic rate (**A**), stomatal conductance (**B**), water use efficiency (**C**), and transpiration rate (**D**) in leaves of cucumber seedlings under drought stress. Different lowercase letters represent significant differences between treatments at the 0.05 level. Values were the means ± SD (n = 3).

*3.3. ROS and MDA Contents*

Except for the $Na_2WO_4$ treatment, e[$CO_2$] reduced superoxide $O_2^-$ production rate and MDA contents compared with a[$CO_2$] treatment (Figure 3A,C). Under the a[$CO_2$], compared with the control treatment, ABA treatment significantly reduced the production rate of $O_2^-$, the content of $H_2O_2$ and MDA by 26%, 29%, and 13%, respectively. $O_2^-$ production rate, $H_2O_2$, and MDA content of $Na_2WO_4$ + ABA treatment were lower than those of $Na_2WO_4$ treatment. In contrast, plants with the $Na_2WO_4$ treatment had the highest rate of $O_2^-$ production, $H_2O_2$, and MDA contents compared to other treatments, both in a[$CO_2$] and e[$CO_2$]. Compared with the control treatment, $Na_2WO_4$ significantly increased the $O_2^-$ production rate, $H_2O_2$, and MDA contents by 16%, 12%, and 47%, respectively (Figure 3).

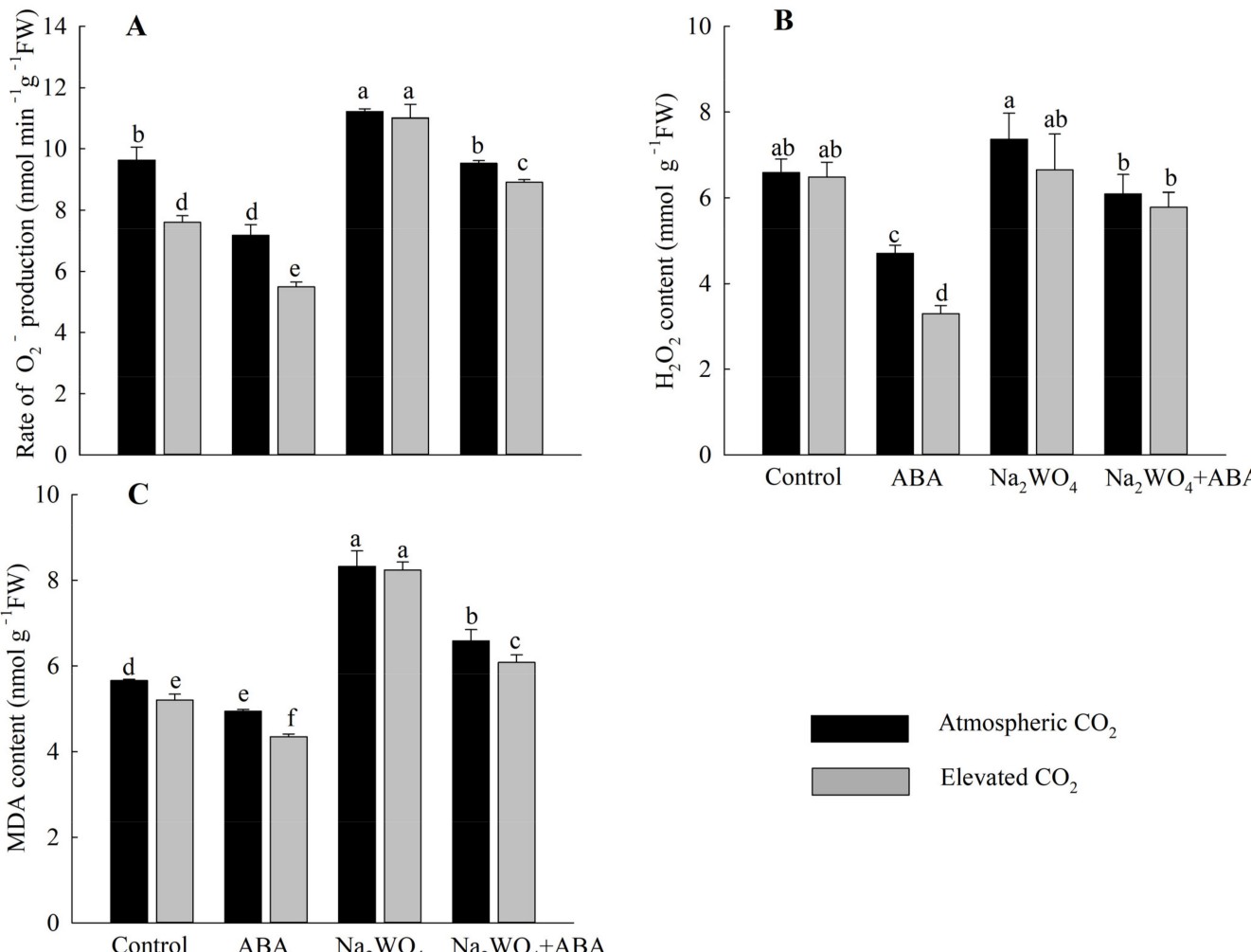

**Figure 3.** Effects of elevated $CO_2$ and exogenous ABA on $O_2^-$ production rate (**A**), $H_2O_2$ content (**B**), and MDA content (**C**) in leaves of cucumber seedlings under drought stress. Different lowercase letters represent significant differences between treatments at the 0.05 level. Values were the means $\pm$ SD (n = 3).

*3.4. Osmoregulation Substances' Contents*

As shown in Figure 4, at the a[$CO_2$] level, the contents of proline, free amino acids, soluble sugar, and soluble protein in the leaves of ABA-treated seedlings under drought stress were significantly higher than those of other treatments. The contents of proline, free amino acids, soluble sugar, and soluble protein in the ABA treatment significantly increased by 32%, 19%, 40%, and 8%, respectively, compared with the control treatment. $Na_2WO_4$ treatment had the lowest contents of each substance, which decreased by 21%, 57%, 23%, and 14% compared with the control treatment. Furthermore, e[$CO_2$] further promoted the accumulation of osmoregulation substances in seedlings, and the differences were significant under ABA treatment. Compared with a[$CO_2$], the contents of proline, free amino acid, and soluble sugar increased by 9%, 23%, and 12%, respectively.

*3.5. Antioxidant Enzyme Activities, ASA, DHA Contents and ASA/DHA Ratio*

It can be observed from Figure 5 that under a[$CO_2$], the activities of SOD, POD, and CAT in leaves of cucumber seedlings treated with ABA were the highest, significantly increased by 11%, 73%, and 10%, respectively, compared with the control treatment. Compared with the control treatment, three enzyme activities of $Na_2WO_4$ treatment were decreased by 18.88%, 9.8%, and 22.72%. The enzyme activities under $Na_2WO_4$ + ABA treat-

ment were higher than those under $Na_2WO_4$ treatment, which indicated that exogenous ABA could weaken the inhibition of sodium tungstate on antioxidant enzymes. Compared with $a[CO_2]$, $e[CO_2]$ further significantly increased the activities of SOD and POD in all treatments. Under $e[CO_2]$, compared with the control treatment, ABA treatment significantly increased the activities of SOD, POD, and CAT (Figure 5A–C).

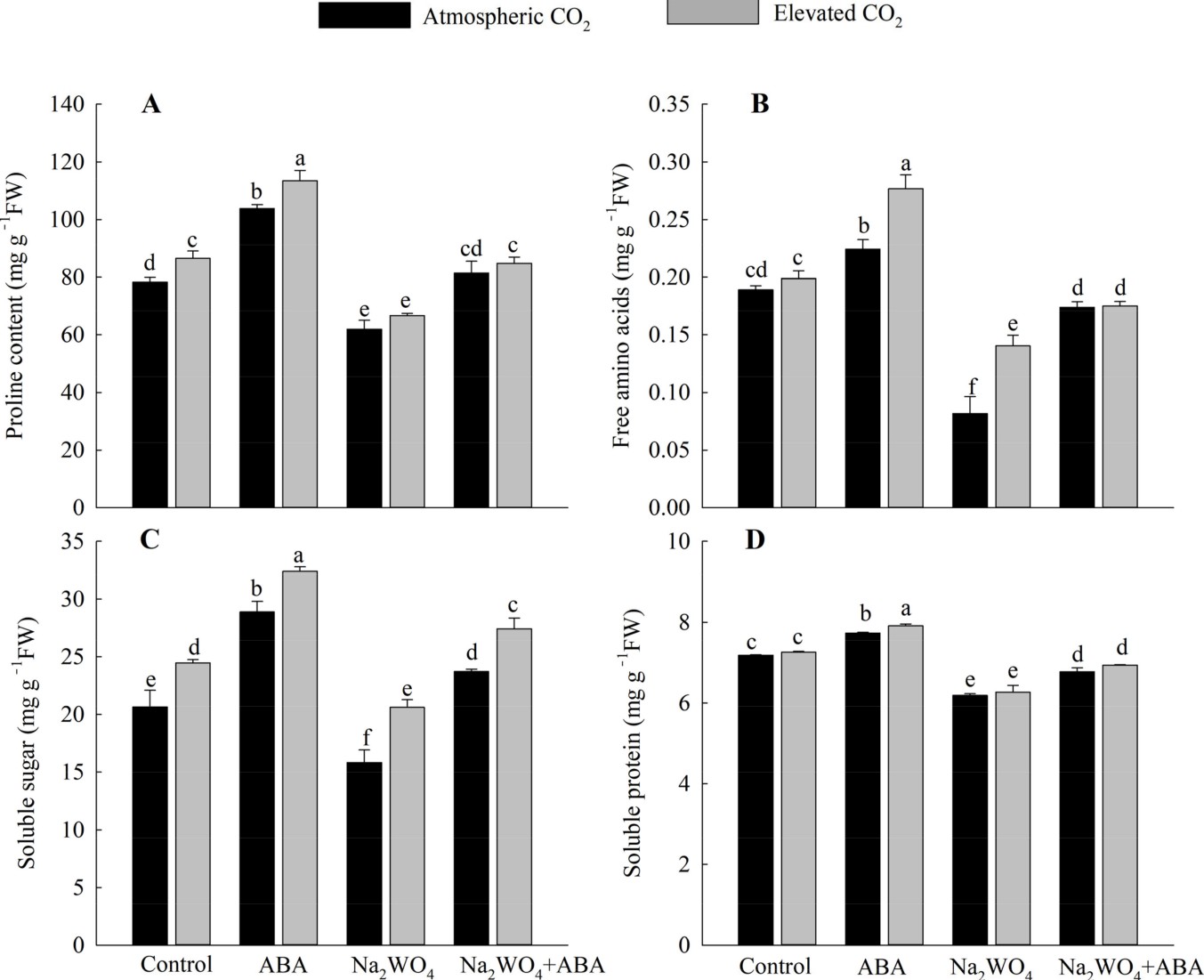

**Figure 4.** Effects of elevated $CO_2$ and exogenous ABA on proline content (**A**), free amino acids (**B**), soluble sugar (**C**), and soluble protein (**D**) in leaves of cucumber seedlings under drought stress. Different lowercase letters represent significant differences between treatments at the 0.05 level. Values were the means $\pm$ SD (n = 3).

It is noticeable that at $a[CO_2]$, the content of ASA and the ratio of ASA/DHA were the highest and the content of DHA was the lowest in ABA treatment while being opposite with $Na_2WO_4$ treatment (Figure 5D,F). The content of ASA and the ratio of ASA/DHA under $Na_2WO_4$ + ABA treatment were increased by 58% and 115%, respectively, and the content of DHA was decreased by 23% compared with $Na_2WO_4$ treatment. Compared with $a[CO_2]$, $e[CO_2]$ could further increase the content of ASA and the ratio of ASA/DHA in cucumber seedling leaves and the differences were significant except for $Na_2WO_4$ treatment, and the content of DHA decreased significantly under $e[CO_2]$ (Figure 5D–F).

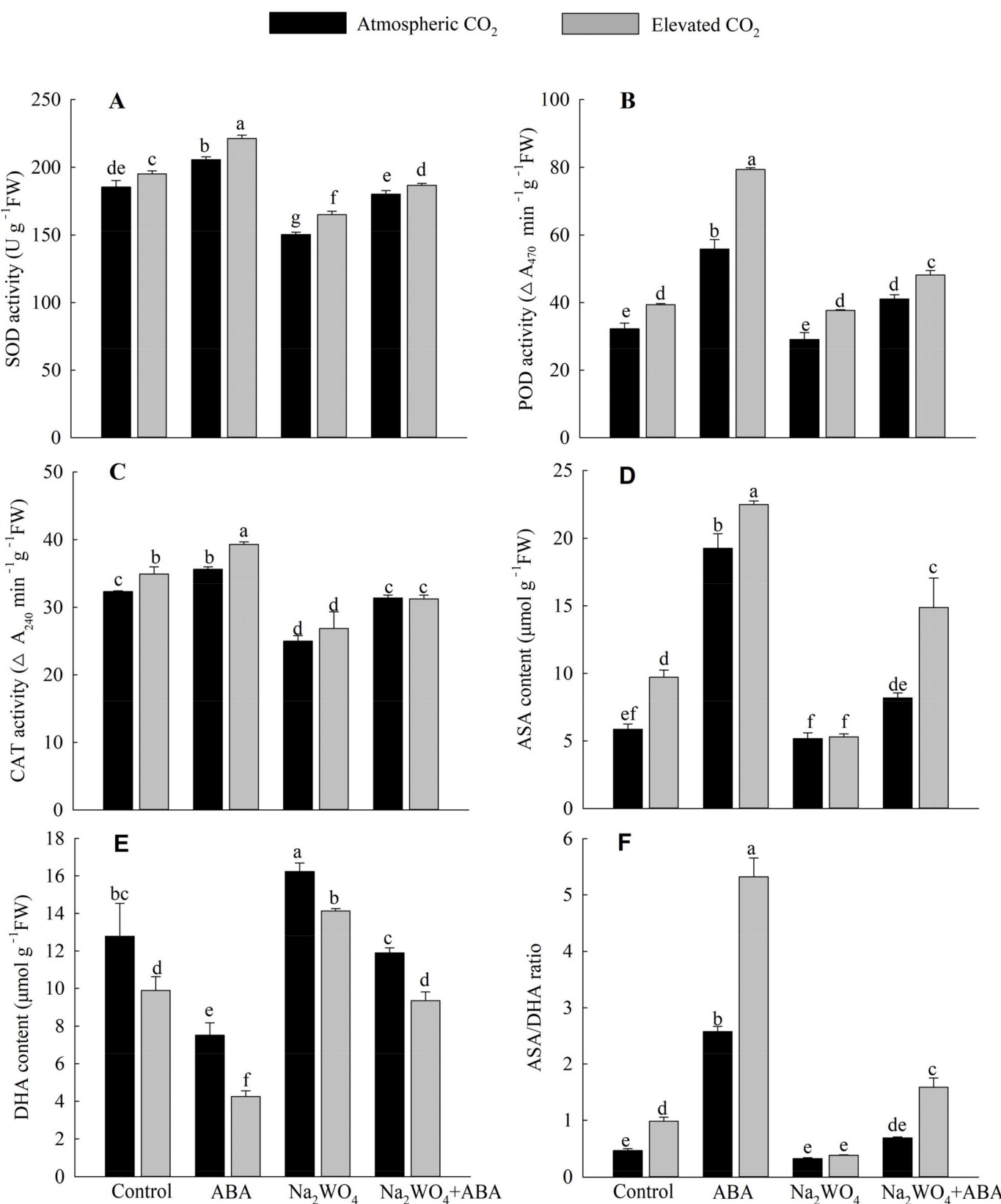

**Figure 5.** Effects of elevated $CO_2$ and exogenous ABA on SOD activity (**A**), POD activity (**B**), CAT activity (**C**), ASA content (**D**), DHA content (**E**), and ASA/DHA ratio (**F**) in leaves of cucumber seedlings under drought stress. Different lowercase letters represent significant differences between treatments at the 0.05 level. Values were the means ± SD (n = 3).

*3.6. GSH, GSSG Contents, GSH/GSSG Ratio and Ascorbate–Glutathione Cycle Enzyme Activities*

Under a[CO₂], the ABA treatment had the highest GSH content and GSH/GSSH ratio, and the lowest GSSG content compared to the other treatments, while the opposite was observed with the Na₂WO₄ treatment (Figure 6A–C). The content of GSH and the ratio of GSH/GSSG of Na₂WO₄ + ABA treatment were increased by 13% and 90%, respectively, and the content of GSSG was decreased by 41% compared with Na₂WO₄ treatment. Compared with a[CO₂], e[CO₂] increased the ratio of GSH/GSSG in cucumber seedling leaves and the differences were significant except for ABA treatment (Figure 6C).

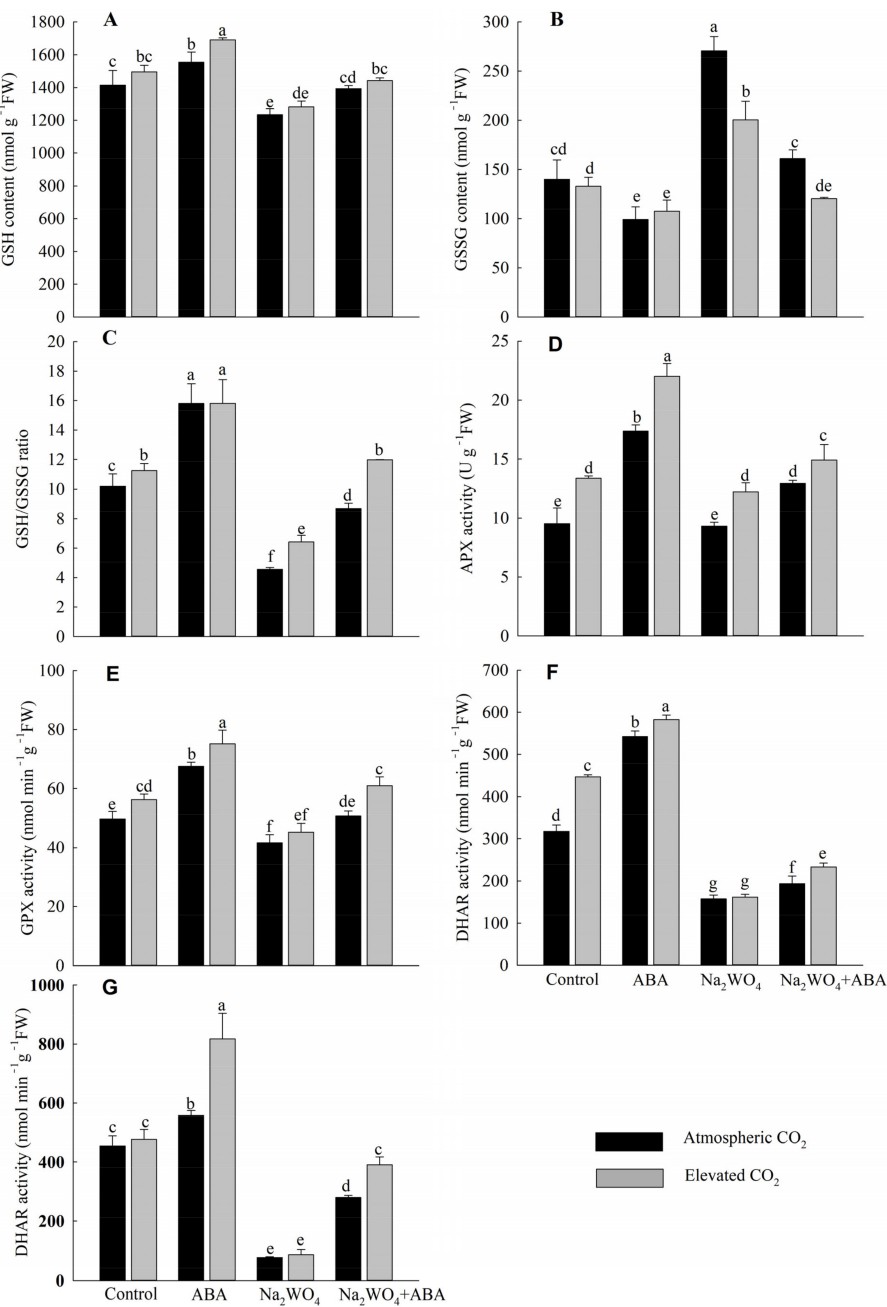

**Figure 6.** Effects of elevated $CO_2$ and exogenous ABA on GSH content (**A**), GSSG content (**B**), GSH/GSSG ratio (**C**), APX activity (**D**), GPX activity (**E**), DHAR activity (**F**), and GR activity (**G**) in leaves of cucumber seedlings under drought stress. Different lowercase letters represent significant differences between treatments at the 0.05 level. Values were the means ± SD (n = 3).

Compared with the control treatment, ABA significantly increased the activities of APX, GPX, DHAR, and GR in the leaves of cucumber seedlings under a[$CO_2$] by 82%, 36%, 71%, and 23%, respectively (Figure 6D–G). While $Na_2WO_4$ treatment decreased them by 2%, 16%, 50%, and 83%, respectively, compared with the control treatment, the differences between DHAR and GR were significant (Figure 6F,G). The enzyme activities under $Na_2WO_4$ + ABA treatment were significantly higher than those under $Na_2WO_4$ treatment. Except for $Na_2WO_4$ treatment, e[$CO_2$] enhanced the activities of APX, GPX, and DHAR significantly, while e[$CO_2$] significantly enhanced GR activity only under ABA and $Na_2WO_4$ + ABA treatments (Figure 6D–G).

## 4. Discussion

Drought stress is one of the largest and most persistent constraints in agricultural development [53]. The cucumber is a major horticultural species, which is frequently used as a model plant to explore plant processes in response to drought stress [54]. At present, other evidence has already indicated that e[$CO_2$] and exogenous ABA can alleviate drought stress in cucumbers [53,55]. However, most studies are aimed at a single effect of either e[$CO_2$] or exogenous ABA spraying for the alleviation of drought stress. $CO_2$ and ABA signals have a variety of interactions on the drought response in plants, which can work orchestrally to allow plants adapt to adverse environments [56]. In this study, we investigated the growth, photosynthetic capacity, active osmoregulation, and antioxidant enzyme activities of cucumber seedlings under e[$CO_2$] and/or exogenous ABA to assess the effects of interaction between e[$CO_2$] and/or exogenous ABA on the drought resistance of cucumber seedlings.

There is a common consensus that drought stress has a significant inhibitory effect on plant growth, in particular on biomass. e[$CO_2$] provides the opportunity to promote the growth of drought-stressed plants and increase plant biomass accumulation [57]. Additionally, ABA can also act as an emergency stress hormone and play a pivotal role in signal transduction under drought stress, thus slowing down the damage to plants [58]. The results of this experiment showed that e[$CO_2$] significantly enhanced the growth of cucumber seedlings under drought stress (Figure 1). On top of that, the application of ABA was more beneficial to plant growth (Table 1). These discoveries suggest that e[$CO_2$] and exogenous ABA positively enhanced the tolerance of cucumber seedlings to drought stress. Moreover, its enhanced drought tolerance might be due to e[$CO_2$] induced stomatal closure and reduced $G_s$, and consequently a decreased water dissipation [59], and this effect is simultaneously enhanced by exogenous ABA [60].

The photosynthetic pigment content is one of the most important physiological traits to determine the photosynthetic capacity and plant growth level of the plant [61]. The results of this experiment showed that e[$CO_2$] and exogenous application of ABA significantly increased the photosynthetic pigment content in leaves of cucumber exposed to drought stress (Table 2), probably because e[$CO_2$] and ABA can reduce the content of ROS in leaves (Figure 3), thus reducing the degradation of chlorophyll by ROS [62]. The increase in chlorophyll content is also a crucial factor that enables the increase in photosynthetic rate in cucumber. Photosynthesis is the most fundamental physiological activity of plants, involving light energy transfer, carbon fixation, $CO_2$ uptake, and $O_2$ release and other related processes, and also plays a key role in balancing the carbon cycle of land ecosystems and is responsible for climate change [63]. In addition, $CO_2$, as a raw material for photosynthesis, has a significant contribution to the effects of photosynthesis, especially for C3 plants [64]. In our study, e[$CO_2$] significantly improved the photosynthetic gas exchange parameters of cucumbers under drought stress (Figure 2). The stomatal conductance and transpiration rate of cucumber leaves were reduced under the e[$CO_2$], but the net photosynthetic rate of cucumber leaves did not decrease (Figure 2). This is probably because the increase in $CO_2$ concentration enhanced the difference in $CO_2$ concentration between the external environment and the leaf pulp cells, which improved the $CO_2$ diffusion dynamics, and the $CO_2$ from the environment could still enter the leaves; therefore, even though there is no

significant change in the intercellular $CO_2$ concentration, current results are still consistent with the findings of Zheng et al. [63]. Similarly, abiotic stresses such as drought can cause damage to the photosynthetic organs of plants, and ABA can alleviate this damage while increasing the content of photosynthetic pigments under stress (Table 2), enhancing the stability of the cystoid membrane, improving the light energy capture ability, strengthening the photochemical reactions of leaves, and protecting the photosynthetic machinery, thus ultimately increasing the photosynthetic rate [65], which is consistent with the results of this experiment.

Drought stress is an important factor limiting the growth and development of plants [66], which can induce the generation of ROS such as $O_2^-$ and $H_2O_2$. Excess ROS can damage cell membranes, proteins, RNA and DNA molecules, reduce photosynthetic efficiency and lead to leaf wilting [67]. To alleviate the damage from ROS, plants evolved a sophisticated antioxidant system to maintain a dynamic balance between the production and clearance of ROS in stressful environments [68]. Previous studies have shown that exogenous spraying of ABA can improve the activities of antioxidant enzymes SOD, POD, and CAT, which reduces the content of MDA and the accumulation of ROS in maize and kiwifruit under drought stress [35,69]. Similar results were observed in our experiment; ABA treatment reduced $O_2^-$ production rate, MDA and $H_2O_2$ content and increased POD SOD, and CAT activities when compared to the control treatment under drought stress (Figures 3 and 5). Nevertheless, after spraying the ABA inhibitor $Na_2WO_4$, both $O_2^-$ production rate, $H_2O_2$, and MDA content were significantly increased compared with the control treatment. These results inferred that exogenous ABA can increase the activities of antioxidant enzymes to eliminate the accumulation of ROS, reduce the degree of membrane lipid peroxidation and protect the integrity of membrane structure, reduce the damage of drought stress on the cucumber seedling biofilm system, and enhance its antioxidant capacity, which was inconsistent with previous studies of cotton [70]. e[$CO_2$] further increased the activities of antioxidant enzymes and decreased the contents of ROS and MDA under drought stress [71,72], which were observed in our results (Figures 3 and 5). The above observation could be explained by the e[$CO_2$] promoting photosynthetic carbon assimilation and inhibiting photorespiration, thus reducing ROS accumulation and $H_2O_2$ production, limiting the Mehler reaction by increasing the utilization rate of NADP+ in photosystem I (PSI) and producing more reducing coenzyme II (NADPH) for the ASA–GSH cycle, which finally induces antioxidant enzyme activity to reduce the damage of drought stress to cucumber seedlings [73,74]. Under the treatment of $Na_2WO_4$, the mitigation ability of $CO_2$ on cucumber seedlings under drought stress was weakened. For example, there were no significant differences between $O_2^-$ production rate, $H_2O_2$, and MDA contents under e[$CO_2$] and a[$CO_2$] (Figure 3), which might be related to the ABA level, implying that exogenous ABA signaling may be involved in regulating plant growth and oxidative stress response to e[$CO_2$]. However, the specific mechanism is not clear yet and needs further study.

Osmoregulation is an important way for plants to resist drought stress. It means that plants actively accumulate organic solutes or inorganic ions to reduce osmotic potential under water stress, to promote plant cells to absorb water from the outside, and to guarantee their physiological processes function normally [75,76]. ABA is a stress hormone and studies have shown that exogenous spraying of ABA can improve the drought tolerance of plants by maintaining water balance, inducing the antioxidant system, maintaining membrane stability, and improving the level of osmotic adjustment substances such as soluble sugar, soluble protein, and amino acids [77,78]. It was reported that exogenous spraying of ABA increased the content of soluble sugar and soluble protein and reduced the damage degree of *Axonopus compressus* under drought stress [78], which is consistent with our results (Figure 4C,D). Proline would accumulate first during drought stress to maintain the osmotic balance between the cell matrix and the environment in wheat and papaya [79–81]. Our results showed that exogenous ABA could increase the content of proline and free amino acids in cucumber seedling leaves (Figure 4A,B). Moreover, e[$CO_2$]



can promote the accumulation of osmotic adjustment substances such as soluble sugar, soluble protein, proline, and free amino acids in cucumber seedlings (Figure 4). When cucumber seedlings were subjected to drought stress, e[$CO_2$] and exogenous ABA could synergistically increase the accumulation of proline and other osmolytes, the osmotic adjustment ability of seedlings, and then their drought resistance (Figure 4). These results indicated that exogenous ABA interacts with e[$CO_2$] on the regulation of plant growth in drought conditions.

In addition to osmoregulation, the ratios of ascorbic acid/dehydroascorbic acid (ASA/DHA), and glutathione/oxidized glutathione (GSH/GSSG) could determine the stress level of plants and reflect the redox status of plants through the ASA–GSH cycle [12,17]. In the present study, we observed that the content of ASA, GSH, and the ratio of ASA/DHA as well as GSH/GSSG in cucumber seedlings were significantly increased after application of e[$CO_2$] and exogenous ABA under drought stress (Figures 5D,F and 6A,C), which is comparable to the studies of Jiang et al. [82] and Liu et al. [83]. These results implied that e[$CO_2$] and ABA treatment are more favorable to provide a reduced state environment for plants, thus improving the antioxidant defense function of plants. The activity of the ASA–GSH cycle mainly depends on the activities of ascorbate peroxidase (APX), glutathione reductase (GR), glutathione peroxidase (GPX), and dehydroascorbate reductase (DHAR). Among them, APX directly reduces $H_2O_2$ to water using ascorbic acid as an electron donor [5]. The results from the present study showed that both ABA treatment and e[$CO_2$] treatment can separately increase the activity of APX, and their interaction enlarged this effect (Figure 6D), indicating that $CO_2$ and ABA can improve the ability to remove active oxygen in cucumber seedlings. DHAR can participate in the regeneration of ASA using GSH as substrate [84], and converting GSH to GSSG through GR [85]. In this pathway, ascorbate and GSH are not consumed in the cycling transfer of electrons but participate in the cycling transfer of reduction equivalent, thus eventually reducing $H_2O_2$ to $H_2O$ [7,85]. In this study, treatment with ABA and e[$CO_2$] increased the activities of DHAR, GPX, and GR, the regeneration of ASA and GSH, and the activity as well as the operation of the whole ASA–GSH cycle system, thus it accelerated the activity clearance and alleviated the oxidative damage of cucumber seedlings under drought stress (Figures 5 and 6). However, interestingly, after the exogenous application of the ABA synthesis inhibitor $Na_2WO_4$, the activities of DHAR, GPX, and GR were not significantly improved by e[$CO_2$]. We speculated that exogenous ABA may be involved in the response of the plant ASA–GSH cycle to e[$CO_2$] under drought stress as an important substance.

## 5. Conclusions

e[$CO_2$] and exogenous ABA synergistically enhanced cucumber seedling drought tolerance by maintaining high photosynthetic capacity, increasing the activity of antioxidant enzymes, and accelerating the ASA–GSH cycle to alleviate the drought-induced oxidative damage in cucumber seedlings (Figure 7). Therefore, the combination of e[$CO_2$] and exogenous ABA might be an effective way to increase crop yield, especially in arid and semi-arid regions. Furthermore, our results also indicated that ABA seemed to be involved in e[$CO_2$]-induced tolerance to drought stress in cucumber seedlings. However, the mechanism of the ABA-mediated $CO_2$ enhancement of drought resistance is still unclear. Future studies of cell membrane structure, intercellular and intracellular water transport-related proteins may shed light on the mechanisms of e[$CO_2$]-induced drought tolerance.

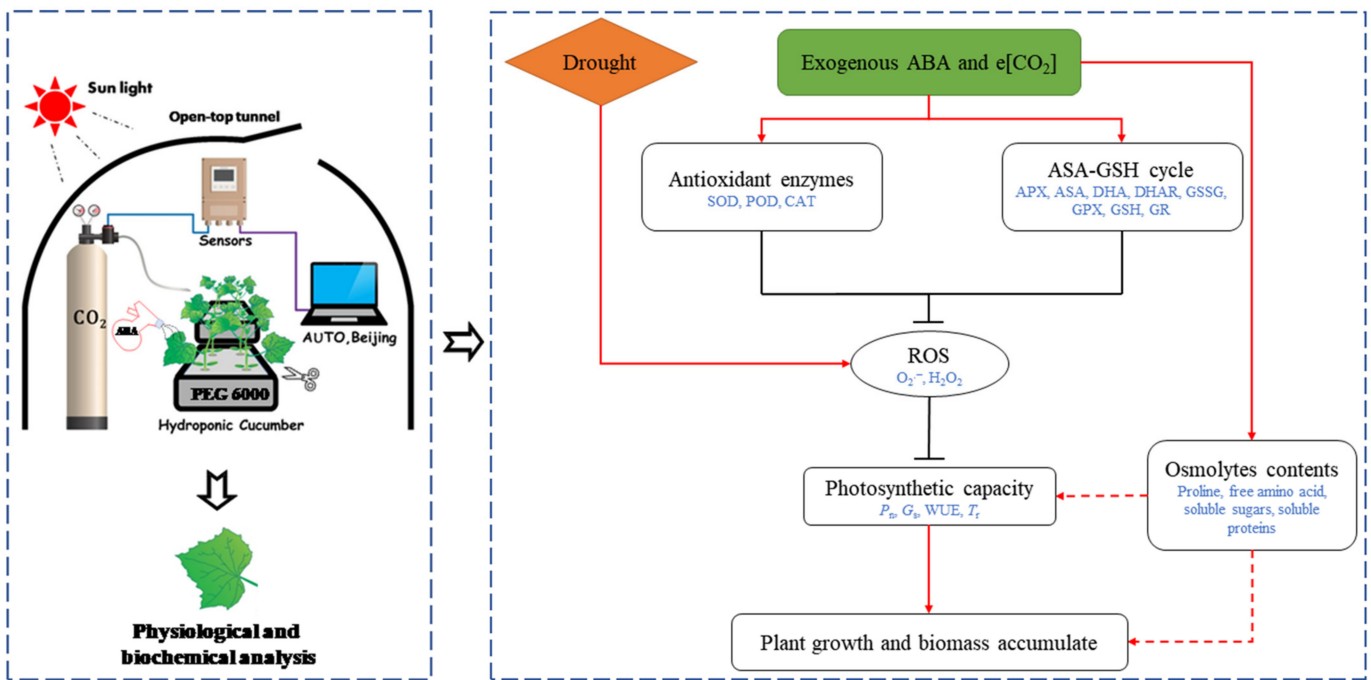

**Figure 7.** Schematic diagram of elevated $CO_2$ and exogenous ABA enhancement of drought resistance of cucumber seedlings. Arrow and bar-ended lines represent activation and inhibition, respectively. Dotted lines denote indirect regulation.

**Author Contributions:** Conceptualization, Z.B. and Q.L.; investigation, Q.S., X.H., T.W. and H.Q.; writing—original draft preparation, X.H., Q.S.; writing—review and editing, X.Y. and Y.C.; supervision, project administration, and funding acquisition, Z.B. and Q.L. All authors have read and agreed to the published version of the manuscript.

**Funding:** This work was supported by the National Natural Science Foundation of China (31872154, 31902092); the Agricultural Science and Technology Innovation Project of the Chinese Academy of Agricultural Sciences (34-IUA-03).

**Data Availability Statement:** The data presented in this study are available upon request from the corresponding author.

**Conflicts of Interest:** The authors declare no conflict of interest.

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
