# Peer review of "The Beneficial Roles of Elevated [CO2] on Exogenous ABA-Enhanced Drought Tolerance of Cucumber Seedlings"

_horticulturae, doi:10.3390/horticulturae9040421_

Round 1

Reviewer 1 Report

The experiment is important and the results are good, with a sufficient number of samples and replicates.  It is well designed. However, there are important questions and points that the authors should clarify. The major flaw of the ms. is that plants water status were no determined.  Although,  water deficit  was  induce  with 5% polyethylene glycol 6000 (PEG 6000, ψ w = -0.05 MPa), but not leaf water content, water potential nor leaf relative water content was measured, so the water status of plant grown in each treatments  was unknown.  Can the authors provide information on any parameters of the plants water status?

Besides, were all plants water stressed with polyethylene glycol? Were there no control plants without polyethylene glycol? If so, then there is no real control of the experiment. What the authors point out as plants subjected to H2O treatment  reality are plants subjected to water deficit (WD), so it should be clarified why there are plants without polyethylene glycol treatment (no WD).

How long did the experiment with e[CO2] spraying last?  Just only five days? If so it is quite short period.

I suggest the authors change "drought" to "water deficit" throughout the manuscript.

The introduction should be brief, so please summarize it. Authors must give a hypothesis of the work in the manuscript

Check the parameters unit, in particular light intensity (mmol m-2 s-1) Ci (mmol mol-1) and WUE (mmol mol-1), please check all units.

The "H2O" treatment is actually plants subjected to water deficit using %% polyethylene glycol, this is confusing throughout the manuscript. Please this should be clarified. 

Figure 1C shows Ci values, which are never mentioned in the results or throughout the Manuscript. if it is not important, I suggest eliminating it.

 The results section is sometimes confusing and could be improved.

Figures 4 and 5 could be merged into a single figure; as well as Figures 6 and 7.

Reviewer 2 Report

Manuscript The beneficial roles of elevated [CO2] on exogenous ABA-enhanced drought tolerance of cucumber seedlings by authors Qiying Sun, Xinrui He, Tengqi Wang, Hengshan Qin, Xin Yuan, Yunke Chen, Zhonghua Bian, Qingming Li is an experimental research related to the use of greenhouse cleanings with carbon dioxide, ABA and sodium tungstate. This study may be of interest as a practical application but contains a small amount of new data of a fundamental nature. The main problem of this study is the lack of understanding of the mechanisms of the relationship between physical factors and the consequences caused by their modulation in plant cells and water quality.

So in the abstract and introduction there are general arguments about the fact that Drought and CO2 concentration are related, but it is not clear how this can be related to this particular case of setting up an experiment. The authors do not consider that carbon dioxide dissolves in water, causing the appearance of the corresponding acid. There is also no explicitly reflected connection between the action and the interaction between the studied effectors. Lines 10-12 should be revised or removed.

In the abstract, commonly used phrases look out of place, since they do not reflect the work of the authors and belong to the category of statements of a redundant type that do not carry information. In the introduction there is a fragment related to the description of the experiment, which should also be corrected. In the last paragraph, information is placed on the goal and task, methods and approaches are placed in a special section below. The question or questions should be formulated.

This also becomes the reason for an unobvious and fuzzy conclusion and a scheme that does not add or explain anything, since the physicochemical aspects of the influence of CO2 on photosynthesis, respiration and productivity, and ABA on changes in transport and metabolism and the maintenance of osmotic pressure, were not considered by the authors.

In the materials and methods there is an indication of the manufacturer's instructions (lines 172, 194-195) in such cases, a link to an open source (article or other) is required.

It is not possible to evaluate.

A significant drawback of the work is the insufficient description of the parameters and tasks. It is not clear to me what type of drought the authors simulated (air drought, soil drought, mixed type drought). How drought parameters were controlled. The lack of photographs of plants also exacerbates the situation.

I would like the conclusions to be clearer in terms of their applicability.

Suzhou, China) according to the manufacturer's instructions.

The application of the form a[CO2] and e[CO2] is not very clear.

In addition, the magazine has color illustrations and I recommend changing the black and white graphics.

Round 2

Reviewer 1 Report

The revised version of the manuscript “ Photosynthetic responses and protective mechanisms of cacao seedlings under prolonged drought stress”; improved considerably,

In the revised version of the manuscript “The beneficial roles of elevated [CO2] on exogenous ABA-enhanced drought tolerance of cucumber seedlings, I think that almost all suggestions were taken and the manuscript improved considerably. I consider that with all the changes incorporated in the manuscript and the response to the concerns of the previous reviewers the manuscript improved considerably. I can recommend the manuscript to be published in horticulturae. However, there are several minor points that still need to be corrected:

On figure 2

gs unit should be with these values (100-300) in mmol m-2 s-1 instead of mol m-2 s-1.

WUE unit should be mmol mol-1

Tr unit should be mmol m-2 s-1 (water loss by transpiration in plants is 1000 orders of magnitude greater than photosynthesis)

For an experiment carried out with an e[CO2] of only 5 days in hydroponic conditions, it seems to me an inadequate conclusion. 

“Therefore, the combination of e[CO2]  and exogenous ABA can be an effective way to increase crop yield, especially in arid and  semi-arid regions.” please left out

Reviewer 2 Report

In general, I am satisfied with the changes made. Considering that the manuscript can be printed.

I would like to note that omics and multiomics studies by themselves will not add any understanding to this issue, since it concerns reversible changes in the structure of membranes, intercellular and intracellular transport and changes in the properties of water due to the solubility of gases in individual cellular compartments, which should be investigated. quite different methods. But everything has its time, and this will also become known and understandable.
